# Influence of Nano-Chitosan Loaded with Potassium on Potassium Fractionation in Sandy Soil and Strawberry Productivity and Quality

Shaimaa Hassan Abd-Elrahman [1], Yasser Abd El-Gawad El-Gabry [2], Fadl Abdelhamid Hashem [3], Mohamed F. M. Ibrahim [4,*], Ehab I. El-Hallous [5], Zahid Khorshid Abbas [6], Doaa Bahaa Eldin Darwish [6,7], Nadi Awad Al-Harbi [8], Salem Mesfir Al-Qahtani [8] and Noura Mohamed Taha [9]

1. Soil and Water Department, Faculty of Agriculture, Ain Shams University, Cairo 11241, Egypt
2. Department of Agronomy, Faculty of Agriculture, Ain Shams University, Cairo 11241, Egypt
3. Central Laboratory for Agricultural Climate, Agricultural Research Center, Giza 12411, Egypt
4. Department of Agricultural Botany, Faculty of Agriculture, Ain Shams University, Cairo 11566, Egypt
5. Department of Biology, College of Science, Taif University, P.O. Box 11099, Taif 21944, Saudi Arabia
6. Department of Biology, Faculty of Sciences, University of Tabuk, Tabuk 71491, Saudi Arabia
7. Botany Department, Faculty of Science, Mansoura University, Mansoura 35511, Egypt
8. Biology Department, University College of Tayma, University of Tabuk, Tabuk 47512, Saudi Arabia
9. Department of Horticulture, Faculty of Agriculture, Ain Shams University, Cairo 11241, Egypt
* Correspondence: ibrahim_mfm@agr.asu.edu.eg

**Abstract:** Under sandy soil conditions, increasing the efficiency of potassium (K) fertilizers is considered to be a major limiting factor for improving the productivity and quality of fruit crops. In this context, utilizing nanotechnology has emerged as a novel technique to increase the efficiency of K applications. In our study, two field trials were conducted, in two consecutive seasons (2019/2020 and 2020/2021), to compare the effects of nano-chitosan loaded with K as a foliar treatment with those of conventional soil applications of K on plant growth, yield, and quality of strawberry plants grown in sandy soil. Strawberry plants were treated with 12 different treatments, which were replicated three times in a randomized complete block design in each growing season. Potassium sulfate ($K_2SO_4$, 48% $K_2O$) was applied to the soil at a rate of 150.0 kg acre$^{-1}$ (recommended rate, 100%). Meanwhile, the spraying of nano-chitosan loaded with K was applied at 1000 mg L$^{-1}$ as a control. In addition, $K_2SO_4$ was applied either individually or in combination at the rate of 112.5 or 75.0 kg acre$^{-1}$ with four nano-chitosan-K dosages (250, 500, 750, and 1000 mg L$^{-1}$). After harvesting, soil samples were collected and prepared to determine K fractions. As well, plant samples were collected to determine the vegetative growth parameters and the foliage content of NPK and chlorophyll. Eventually, the yield traits and quality parameters were evaluated. A principal component analysis was conducted to determine the interrelationships of the treatments' averages and their effects on yield components and quality traits. A combined analysis was performed for the two studied seasons and the values were the mean of six replications. The results indicated that the application of common K fertilizer (150.0 kg $K_2SO_4$ acre$^{-1}$) resulted in the maximum increase in soluble and exchangeable K in the soil, which was comparable to those observed with 112.5 kg $K_2SO_4$ acre$^{-1}$ + 1000 mg L$^{-1}$ nano-chitosan-K and 112.5 $K_2SO_4$ acre$^{-1}$ + 750 mg L$^{-1}$ nano-chitosan-K. The total yield, marketable yield, and fruit firmness were all significantly increased by the latter two treatments compared to the control group. Furthermore, plots treated with 112.5 kg $K_2SO_4$ acre$^{-1}$ + 1000 mg L$^{-1}$ nano-chitosan-K significantly increased the total soluble solids, vitamin C levels, acidity, total sugar, and anthocyanin levels in strawberry fruits. In conclusion, under sandy soil conditions, the utilization of nanoparticles could be an indispensable tool for manipulating fertilization management when cultivating strawberries. The K status of the soil was improved by applying 75% of the recommended dose of mineral K in combination with 1000 or 750 mg L$^{-1}$ of nano-chitosan-K, without compromising strawberry yield or quality.

**Keywords:** nanoscale chitosan; nutrient deficiency; nutrient uptake; soil potassium content; strawberry fruit yield

## 1. Introduction

Strawberry is one of the most valuable cash crops for both domestic consumption and export. It is high in vitamins (B9 and C), minerals (P, K, and Ca), and fiber, and it is one of the best natural sources of antioxidants [1]. The use of prohibited chemical fertilizers and pesticides has recently caused environmental pollution in the strawberry cultivation sector, and the untrustworthy results of biological monitoring have been thoroughly investigated [2]. Although that chemical fertilizers increase crop productivity, they produce several hazardous residues that have massive negative impacts on the ecosystem and human health, as well as sustainability losses, and water contamination [3–5].

Innovative products, such as nontoxic fertilizers, should be used to lessen agricultural chemicals' risks and adverse effects. In this regard, nanotechnology has provided useful and distinctive applications in the agricultural industry [6]. Nanofertilizers have been widely used to promote plant growth and to increase crop yields [7]. Nano-engineered elements have been shown to be less toxic than macro- and micro-engineered elements [8]. Nanofertilizers having a size that is less than the pore size of the cell wall, and therefore can easily move through the cell wall and migrate up to the plasma membrane. After gaining entry into the cell, nano-sized elements can move through both apoplastic and symplastic pathways, ultimately affecting plants' various physiological and metabolic activities [9].

Potassium (K) is an essential nutrient for plant growth which influences carbohydrate metabolism, fruit quality, and stress tolerance [10,11]. Potassium in soil is classified into four forms depending on the basis of its availability to plants, i.e., water-soluble K, exchangeable K, K that is difficult to exchange or fix in the lattice structure of clay minerals, and total K [12]. Among these forms, only two forms are available for plant absorption, water-soluble K (that constitutes about 0.1–0.2% of the total amount) which can be absorbed by plants and microbes but is prone to leaching, and exchangeable K (1–2% of the total K in soil) [13]. Furthermore, it is well known that sandy soils are low in nutrients, especially K. Therefore, the addition of K to soil is essential for sustaining plant growth and yields [13]. Compared with traditional fertilizers, nano-K fertilizer for grapes has been shown to significantly increase plants' growth, yields, and nutrient contents [14]. In addition, [9] found that K nanoparticles enhanced wheat's morphological, biochemical, and yield-based parameters.

Nano-chitosan is a chitin-derived biopolymer that improves plant root growth, soil porosity, and water-use efficiency. Chitosan derivatives (e.g., *N*-succinyl and *N*, *O-carboxymethyl* chitosan) have been reported to improve chlorophyll fluorescence and photosynthesis [15]. Chitosan has been shown to regulate gene expression in plant defense pathways, and this has helped to protect fruit for long periods, especially after fruit has been harvested and is in storage [16]. Furthermore, nano-chitosan is a biocompatible and biodegradable material, and composites containing chitosan have performed well without harming naturally occurring beneficial soil microbes [15,17]. In addition, because it improves particle–particle cohesion, nano-chitosan has the potential to be a soil stabilizer [17].

Nanofertilizers assist in minimizing fertilizer waste, reducing environmental contamination, and improving plant nutrient bioavailability [18,19]. According to [20], nano-chitosan-NPK fertilizer increased the growth and productivity of wheat planted in sandy soil. It also acted in soil as a slow-release fertilizer that conserved nutrients from chemical processes such as denitrification, hydrolysis, and leaching that affect the presence of nutrients in soil [20]. To the best of our knowledge, no research has been conducted on the effect of nano-chitosan-K fertilizer on strawberry growth, nutrition, and antioxidant properties. We hypothesized that the combination of K and chitosan would have a compatible and beneficial effect on the yield and quality of strawberry crops. The primary goal of this study was to determine the effect of nano-chitosan-K fertilizer on strawberry plants

growth, yields, nutrient uptake, and juice quality. In addition, in this study, we examined K fractions in cultivated soil.

## 2. Materials and Methods

### 2.1. Study Site Attributes

The field experiment was conducted on strawberry plants (*Fragaria × ananassa* cv. Florida Beauty) during the 2019/2020 and 2020/2021 seasons, at the experimental farm of the Faculty of Agriculture, Ain Shams University, located in Imam Malik village, Kom Hamada, El-Bahira governorate, Egypt (latitude $30°30'36.5''$ N and longitude $30°18'34.3''$ E). The soil used in the experiment had a sandy texture and had been organized into the soil order *Typic Torripsamment* by [21]. Before cultivation, the experimental soil's basic physical and chemical properties were determined (Table 1), using the procedures outlined by [22,23]. The study location was in an arid zone, with an air temperature of $29.3 \pm 7.4$ °C, relative humidity of $55 \pm 7\%$, solar radiation of $20 \pm 5$ MJ m$^{-2}$ day$^{-1}$, and wind speed of $2.5 \pm 0.6$ m s$^{-1}$, as provided by the meteorological station of the Central Laboratory of Agricultural Climate (CLAC) of the Agricultural Research Center, Egypt.

**Table 1.** Properties of the soil (0–30 cm depth) at Imam Malik village, Kom Hamada, El-Bahira governorate, Egypt.

| Parameter | Unit | Value |
|---|---|---|
| Physical properties | | |
| Sand | % | $83.0 \pm 0.2$ |
| Silt | % | $7.30 \pm 0.1$ |
| Clay | % | $9.70 \pm 0.1$ |
| Textural class | - | Sandy |
| Field capacity | % | $17.8 \pm 0.06$ |
| Welting point | % | $6.91 \pm 0.02$ |
| Available water | % | $10.9 \pm 0.04$ |
| Chemical analysis | | |
| Acidity pH (1:2.5) | – | $7.60 \pm 0.1$ |
| Electrical conductivity | dS m$^{-1}$ | $2.30 \pm 0.3$ |
| Organic matter | % | $0.71 \pm 0.02$ |
| Total nitrogen | % | $0.05 \pm 0.02$ |
| Available phosphorus | μg g$^{-1}$ | $12.0 \pm 0.3$ |
| Available potassium | μg g$^{-1}$ | $168 \pm 2.4$ |
| Calcium carbonate | g kg$^{-1}$ | $58.3 \pm 0.4$ |
| Soluble ions | meq L$^{-1}$ | |
| Calcium | | $8.60 \pm 0.4$ |
| Magnesium | | $5.50 \pm 0.1$ |
| Sodium | | $8.50 \pm 0.2$ |
| Potassium | | $0.76 \pm 0.03$ |
| Chloride | | $9.80 \pm 0.2$ |
| Bicarbonate | | $3.95 \pm 0.1$ |
| Sulfate | | $9.61 \pm 0.2$ |

Carbonate ions were not detected. Values were the mean of 6 replicates ± SE.

### 2.2. Experimental Design and Treatments

The strawberry plants were treated with twelve treatments and replicated three times in a randomized complete block design, during each studied season. Control treatments included applying potassium sulfate (K$_2$SO$_4$, 48.0% K$_2$O) in the soil at the rate of 150.0 kg acre$^{-1}$ (recommended rate, 100%) and spraying nano-chitosan-K at the rate of 1000 mg L$^{-1}$. K$_2$SO$_4$ was also applied at rates of 112.5 or 75.0 kg acre$^{-1}$ alone or in combinations with four nano-chitosan-K doses (250, 500, 750, and 1000 mg L$^{-1}$). The studied treatments and their abbreviations are presented in Table 2.

**Table 2.** The studied treatments and their abbreviations.

| | |
|---|---|
| T1 | 150.0 kg $K_2SO_4$ acre$^{-1}$ (control treatment, 100% of the recommended dosage) |
| T2 | 112.5 kg $K_2SO_4$ acre$^{-1}$ + 1000 mg L$^{-1}$ nano-chitosan-K |
| T3 | 112.5 kg $K_2SO_4$ acre$^{-1}$ + 750 mg L$^{-1}$ nano-chitosan-K |
| T4 | 112.5 kg $K_2SO_4$ acre$^{-1}$ + 500 mg L$^{-1}$ nano-chitosan-K |
| T5 | 112.5 kg $K_2SO_4$ acre$^{-1}$ + 250 mg L$^{-1}$ nano-chitosan-K |
| T6 | 112.5 kg $K_2SO_4$ (75% of the recommended dosage) |
| T7 | 75.0 kg $K_2SO_4$ acre$^{-1}$ + 1000 mg L$^{-1}$ nano-chitosan-K |
| T8 | 75.0 kg $K_2SO_4$ acre$^{-1}$ + 750 mg L$^{-1}$ nano-chitosan-K |
| T9 | 75.0 kg $K_2SO_4$ acre$^{-1}$ + 500 mg L$^{-1}$ nano-chitosan-K |
| T10 | 75.0 kg $K_2SO_4$ acre$^{-1}$ + 250 mg L$^{-1}$ nano-chitosan-K |
| T11 | 75.0 kg $K_2SO_4$ (50% of the recommended dosage) |
| T12 | 1000 mg L$^{-1}$ nano-chitosan-K (control treatment) |

Potassium sulfate was applied as a soil application, while nano-chitosan-K was applied as a foliar application. These applications were divided equally and applied at three stages, i.e., at the beginning of vegetative growth (1 week after transplanting, WAT), after the fruit setting stage (6 WATs), and at the fruiting stage (12 WATs). The nano-chitosan-K fertilizer was prepared at the Genetic Engineer Department, Faculty of Agriculture, Ain Shams University, Cairo, Egypt. Low molecular weight chitosan ($\geq$75.0% degree of deacetylation, viscosity from 20 to 300 cPs, average Mw ~50 kDa, product No. 448869) was purchased from Sigma-Aldrich (Saint Louis, MO, USA). Chitosan nanoparticles were prepared using an aqueous solution (50 mg L$^{-1}$), as described by [20], while potassium nanoparticles (K, purity > 99.0, APS 80–100 nm, Mw = 39.0983 g mol$^{-1}$, density 0.862 g cm$^{-3}$, Cat No. NCZ-MN-071/20) were purchased from Nanochemazone (Ave, Leduc, AB, Canada). Certain concentrations of K nanoparticles (250, 500, 750, and 1000 mg L$^{-1}$) were added into the chitosan nanoparticle solution under magnetic stirring for 6 h, at 25 °C. The amount of nutrient loading was examined by checking concentration changes. Transmission electron microscopy (TEM) images of nano-chitosan-K fertilizer are shown in Figure 1.

*2.3. Crop Cultivation and Practices*

The strawberry transplants were acquired from the Agricultural Research Center in the Giza governorate, Egypt. Using the plastic culture management system (drip irrigation and soil mulch), the experiments began on the 10th and 14th of October in 2019/2020 and 2020/2021, respectively, and ended in the middle of April for each season.

Fertilizers were broadcast during soil preparation. The following amounts of fertilizer were added per acre: 20.0 m$^3$ cattle manure, 10.0 m$^3$ chicken manure, 150.0 kg calcium superphosphate, 50.0 kg magnesium sulfate (MgSO$_4$), and 300.0 kg sulfur (recommended by the Egyptian Ministry of Agriculture and Land Reclamation for strawberry cultivation), while 150.0 kg acre$^{-1}$ of ammonium sulfate [(NH$_4$)$_2$SO$_4$] was added in four equal doses during soil preparation, 3, 8, and 12 WATs.

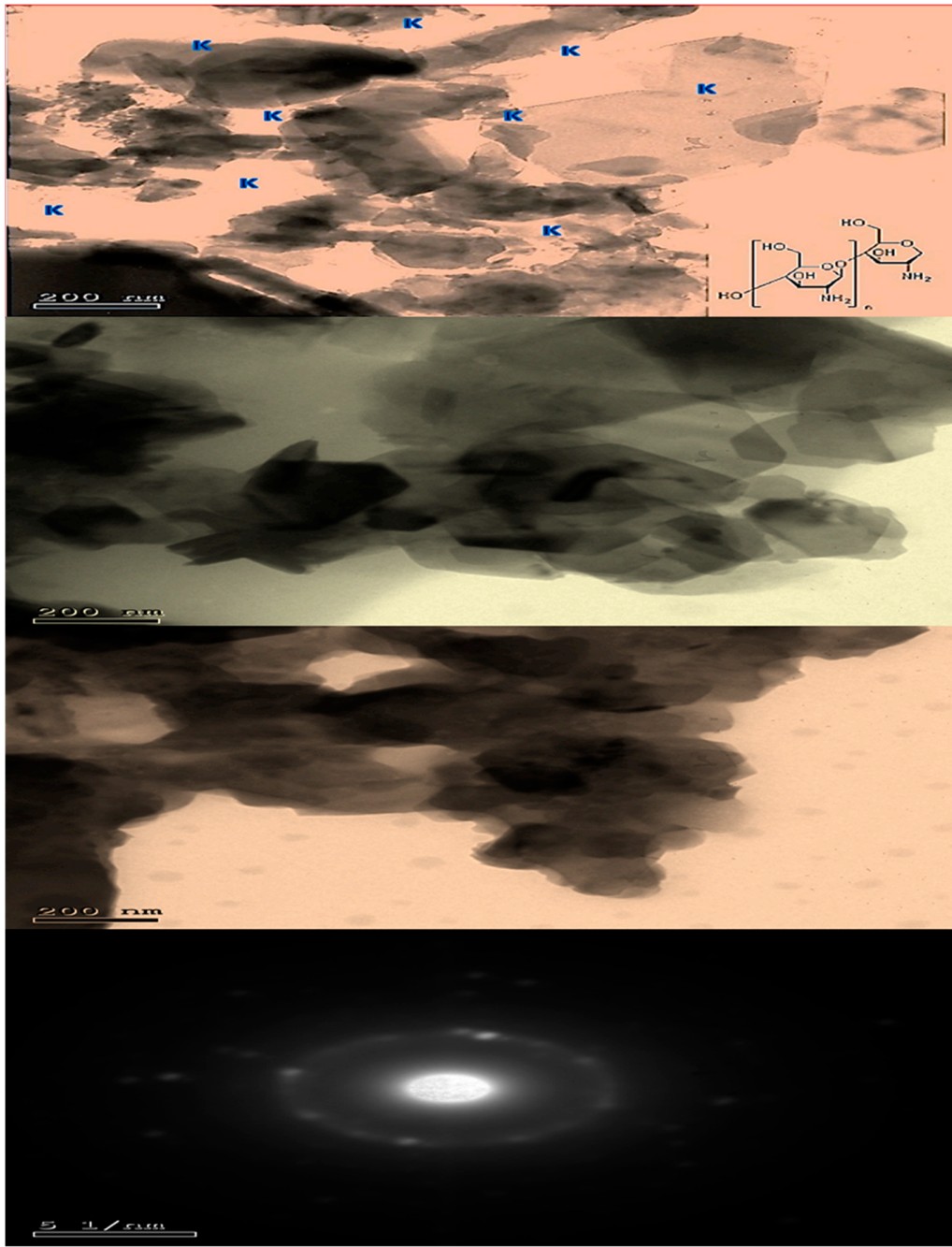

**Figure 1.** Transmission electron microscopy (TEM) images of nano-chitosan loaded with K with size $\leq 200$ nanometers (nm).

The soil was fumigated with $50\,\mathrm{g\,m^{-2}}$ of methyl bromide. The transplants were soaked in a fungicide solution for 20 min before being planted in four rows on each bed, 30 cm apart, with a 25 cm row-to-row spacing. The net plot size was 21 m$^2$, with five ridges (3.5 m in length, 1.2 m in width, and 40 cm in height). When the transplants had the first 2–3 leaves, the beds were covered with 60 micron clear plastic mulch. Transplants were irrigated with overhead sprinklers for two weeks after transplanting to aid plant establishment, and then with drip irrigation tubing emitters until the end of the season. The irrigation water had a pH of 7.04 and an electrical conductivity of $0.72\,\mathrm{dS\,m^{-1}}$. The irrigation system was set up with a control head (media and screen filters, pressure gauges, and control valves). The main line was 75.0 mm diameter PVC pipe with a pressure rating of 6.0 bar and the submain line was 50.0 mm diameter PVC pipe with a pressure rating of 6.0 bar. Lateral lines

were 16.0 mm diameter polyethylene tubes with built-in emitters, 30.0 cm emitter spacing, and a manufacturing emitter discharge of 4.0 L h$^{-1}$, at an operating pressure of 1.0 bar. The amounts of irrigation water were calculated depending on the climatological data for the experimental site to estimate the evapotranspiration using the Penman-Monteith equation, FAO 56 method, presented by [24]. Irrigation water was scheduled on the basis of soil moisture content at field capacity, taking into consideration the climatic conditions, until the end of the experiment. The total amounts of applied irrigation were 2178 and 2209 m$^3$ acre$^{-1}$ during the two studied seasons, respectively.

### 2.4. Data Collection

### 2.4.1. Potassium Fractionation in Soil

After harvesting, soil samples (0–30 cm) were collected from the rhizosphere. The samples were air-dried and crushed to pass through a 2 mm sieve. The crushed samples were examined for the presence of different K fractions. In this respect, water-soluble K ($H_2O$-K) was extracted by shaking 2.5 g of soil in 50 mL of deionized water for 30 min [12]. According to [25], the exchangeable K (EXC-K) was determined using 1.0 M of $NH_4OAC$ for 5 h, at 25 °C. The K that was difficult to exchange or fixed K (FIX-K) was determined using 1.0 M of $HNO_3$, for 2 h, at 100 °C [26]. Total K (TOT-K) was estimated by digesting soil samples with an $H_2SO_4/H_2O_2$ mixture [23].

### 2.4.2. Plant Traits

#### Strawberry Growth

During the two seasons, five plants were harvested from the experimental plot on March 20 and on March 25 to determine plant height (PH), fresh and dry weights plant$^{-1}$ (PFW and PDW, respectively), and leaf number plant$^{-1}$. Moreover, leaf area (LA) was measured using a portable digital leaf area meter (LI-300 area meter produced by LI-COR, Lincoln, Nebraska, USA).

#### Strawberry Foliage Chemical Composition

Plant foliage samples were oven dried at 70 °C until they reached a constant weight. The dry samples were pulverized separately, and a sample was acid digested to determine the total NPK. The total N in the plant was determined using a micro-Kjeldahl apparatus, as described by [27]. Total P was determined colorimetrically, as outlined by [28]. Total K was measured using a Flame photometer, as described by [27]. Furthermore, chlorophyll content was measured using a 502-SPAD device by taking readings of five fresh leaves for each plant in the treatment [29], and the results were expressed in SPAD units [30].

#### Strawberry Yield

Fruits with at least 75% red color were harvested three times per week. Early fruit yield (EFY) was calculated by aggregating pickings up to six weeks after the beginning of the harvesting season. The total fruit yield (TFY) was calculated as the fresh weight of all harvested fruits over the growing season. However, the marketable fruit yield (MFY) was calculated after removing the cull fruit (diseased or miss-shaped). Unmarketable fruit yield included spoiled, malformed, green-shouldered, water-damaged, and rotted fruit. In addition, fruit firmness (FF) was measured using a TA-1000 firmness analyzer instrument with a 1 mm in diameter penetrating cylinder to a constant distance (3 and 5 mm) inside the pulp of fruits at a constant speed of 2 mm s$^{-1}$, and the peak of resistance was recorded as g cm$^{-2}$.

#### Strawberry Fruit Quality

A random sample of ten fully ripe fruits was taken from each experimental plot to determine fruit quality properties using AOAC-described methods [31]. Total soluble solids (TSS) were determined using a digital refractometer (Abbe Leica model). The ascorbic acid content (vitamin C, VC) was calculated as mg 100 g$^{-1}$ fresh weight using

2.6 dichlorophenols indophenol as a titration indicator. Acidity (AC) was determined by titration of fruit juice against 0.1 N NaOH to pH 8.1 and expressed as a percentage of citric acid. The Lane and Eynon method was used to determine the total sugar (TS) in fresh strawberries. A spectrophotometric analysis with HCl (1.5 N) was used to determine the presence of anthocyanin (ANTHO).

*2.5. Statistical Analysis*

Analysis of variance was used to properly analyze the experimental data [32]. Using the Mstat software package, an analysis of variance was performed to statistically analyze data. The variables' means in the combined two seasons were compared using Duncan's multiple range test, with a 5% probability level and a standard error (SE). The principal component analysis (PCA) was conducted using the R statistical software (version 3.6.1) on the investigated treatments' averages to determine their interrelationships and roles in yield components and quality traits.

**3. Results**

*3.1. Potassium Fractionation in Soil*

As shown in Figure 2, the most notable increases in water-soluble K, exchangeable K, and total K in the 2019/2020 and 2020/2021 combined seasons were observed with 150.0 kg $K_2SO_4$ acre$^{-1}$ (T1), 112.5 kg $K_2SO_4$ acre$^{-1}$ + 1000 mg L$^{-1}$ nano-chitosan-K (T2), and 112.5 kg $K_2SO_4$ acre$^{-1}$ + 750 mg L$^{-1}$ nano-chitosan-K (T3). In contrast, the addition of 75.0 kg $K_2SO_4$ acre$^{-1}$ + 250 mg L$^{-1}$ nano-chitosan-K (T10), 75.0 kg $K_2SO_4$, (T11), or 1000 mg L$^{-1}$ nano-chitosan-K (T12) resulted in higher K fixation in the soil in both seasons. The lowest values of fixed K in the soil in both seasons were obtained by T1, T2, and T3.

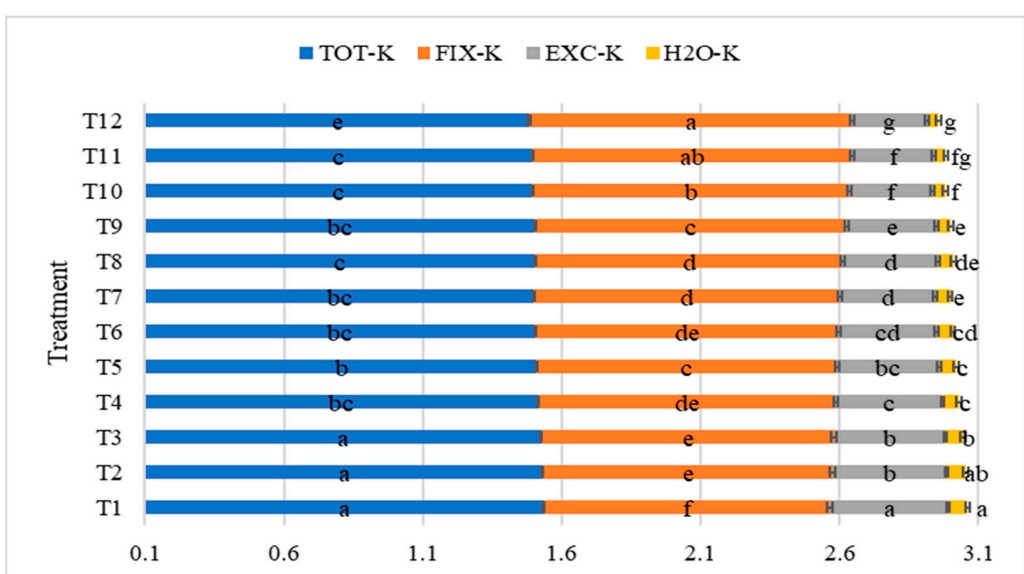

**Figure 2.** Effect of potassium (K) treatments on the percentage of total K (TOT-K), fixed K (FIX-K), exchangeable K (EXC-K), and water-soluble K ($H_2O$-K) in the soil after strawberry harvesting at the combined two studied seasons. Values were the mean of six replicates. Different letters within columns indicate that there were significant differences at a 0.05 level of probability.

*3.2. Plant Traits*

3.2.1. Strawberry Growth

In the combined two studied seasons, the maximum values recorded were for T7 (for plant height), T2 (for plant fresh and dry weights), T2 and T3 (for leaf numbers), and T2 (for leaf areas), as presented in Table 3. Accordingly, as a general observation, the applications of 112.5 kg $K_2SO_4$ acre$^{-1}$ + 1000 mg L$^{-1}$ nano-chitosan-K (T2); 112.5 kg $K_2SO_4$ acre$^{-1}$ +

750 mg L$^{-1}$ nano-chitosan-K (T3), and 75.0 kg K$_2$SO$_4$ acre$^{-1}$ + 1000 mg L$^{-1}$ nano-chitosan-K (T7) were the important practices (equaling the traditional practice with T1 of 150.0 kg K$_2$SO$_4$ acre$^{-1}$) for enhancing strawberry growth and development. In contrast, adding 75.0 kg K$_2$SO$_4$ as a ground application (T11) or 1000 mg L$^{-1}$ nano-chitosan-K as a foliar application (T12) resulted in lower values of the studied vegetative growth parameters than the two treatments combined.

**Table 3.** Effect of potassium treatments on vegetative growth characteristics of strawberry plants in the combined seasons of 2019/2020 and 2020/2021.

| Treatment | Plant Height (cm) | Fresh Weight (g Plant$^{-1}$) | Dry Weight (g Plant$^{-1}$) | Number of Leaves Plant$^{-1}$ | Leaf Area (cm$^2$) |
|---|---|---|---|---|---|
| T1 | 20.14 ± 0.45 c | 101.41 ± 1.12 c | 34.30 ± 0.41 c | 24.58 ± 0.71 b | 566.22 ± 4.86 b |
| T2 | 23.32 ± 0.18 b | 111.67 ± 0.22 a | 46.76 ± 0.59 a | 26.90 ± 0.87 a | 618.88 ± 5.74 a |
| T3 | 22.74 ± 0.20 b | 105.04 ± 0.67 b | 38.62 ± 1.15 b | 27.15 ± 0.32 a | 602.07 ± 1.89 ab |
| T4 | 20.50 ± 0.20 c | 98.63 ± 0.26 d | 28.47 ± 0.10 e | 21.16 ± 0.67 c | 442.32 ± 0.92 d |
| T5 | 19.81 ± 0.24 c | 98.13 ± 0.89 d | 26.70 ± 0.11 f | 20.84 ± 0.48 c | 490.73 ± 52.4 c |
| T6 | 18.47 ± 0.23 d | 80.29 ± 0.53 g | 20.53 ± 0.76 i | 16.83 ± 0.27 de | 386.20 ± 17.2 e |
| T7 | 24.67 ± 0.48 a | 101.91 ± 0.13 c | 32.14 ± 1.32 d | 25.68 ± 0.58 ab | 501.86 ± 1.74 c |
| T8 | 19.77 ± 0.47 c | 93.26 ± 0.83 e | 24.23 ± 0.56 g | 20.17 ± 0.19 c | 412.73 ± 1.40 de |
| T9 | 19.82 ± 0.03 c | 93.58 ± 0.16 e | 23.06 ± 0.24 gh | 18.25 ± 0.19 d | 409.06 ± 2.98 de |
| T10 | 18.88 ± 0.31 d | 88.07 ± 0.80 f | 21.50 ± 0.83 hi | 17.70 ± 0.25 de | 370.28 ± 2.01 ef |
| T11 | 17.39 ± 0.14 e | 77.39 ± 0.17 h | 17.34 ± 0.57 j | 16.40 ± 0.58 e | 330.33 ± 4.91 f |
| T12 | 20.00 ± 0.33 c | 92.37 ± 0.60 e | 21.49 ± 0.94 hi | 18.35 ± 0.35 d | 396.09 ± 3.97 de |

The values were the means of six replicates ± standard errors. Different letters within columns indicate that there were significant differences at a 0.05 level of probability.

### 3.2.2. Strawberry Foliage Chemical Composition

The T2 and T3 treatments (for N content), T2 treatment (for P and K contents), T2 and T3 treatments (for chlorophyll content) were the most effective in improving the chemical composition of the strawberry foliage in the combined two studied seasons (Table 4). Briefly, in the two seasons, the chemical composition of the strawberry foliage responded highly positively to the applications of 112.5 kg K$_2$SO$_4$ acre$^{-1}$ + 1000 mg L$^{-1}$ nano-chitosan-K (T2) and 112.5 kg K$_2$SO$_4$ acre$^{-1}$ + 750 mg L$^{-1}$ nano-chitosan-K (T3). During the two seasons, the T11 treatment produced strawberry plant foliage with the lowest levels of NPK and chlorophyll.

**Table 4.** Effect of potassium treatments on nitrogen, phosphorus, potassium, and chlorophyll contents of strawberry plant foliage in the combined seasons of 2019/20 and 2020/21.

| Treatment | Nitrogen, % | Phosphorus, % | Potassium, % | Chlorophyll, SPAD |
|---|---|---|---|---|
| T1 | 2.02 ± 0.14 def | 0.54 ± 0.0 c | 1.75 ± 0.01 c | 45.70 ± 0.49 b |
| T2 | 2.58 ± 0.07 a | 0.59 ± 0.01 a | 1.86 ± 0.01 a | 48.17 ± 0.20 a |
| T3 | 2.56 ± 0.05 a | 0.56 ± 0.01 b | 1.80 ± 0.01 b | 47.31 ± 0.47 ab |
| T4 | 2.26 ± 0.02 b | 0.51 ± 0.0 d | 1.58 ± 0.01 e | 40.61 ± 0.29 c |
| T5 | 2.21 ± 0.02 bc | 0.51 ± 0.0 d | 1.55 ± 0.01 e | 40.76 ± 0.87 c |
| T6 | 1.91 ± 0.01 f | 0.43 ± 0.0 h | 1.35 ± 0.01 h | 36.09 ± 0.39 e |
| T7 | 2.50 ± 0.01 a | 0.52 ± 0.01 d | 1.64 ± 0.0 d | 45.56 ± 0.88 b |
| T8 | 2.24 ± 0.04 bc | 0.49 ± 0.01 e | 1.48 ± 0.01 f | 38.16 ± 1.10 d |
| T9 | 2.11 ± 0.03 bcd | 0.46 ± 0.01 fg | 1.47 ± 0.03 f | 37.61 ± 0.11 de |
| T10 | 2.02 ± 0.02 def | 0.45 ± 0.01 g | 1.39 ± 0.02 g | 37.59 ± 0.66 de |
| T11 | 1.94 ± 0.04 ef | 0.44 ± 0.0 h | 1.18 ± 0.02 i | 32.27 ± 0.36 f |
| T12 | 2.08 ± 0.02 cde | 0.47 ± 0.0 f | 1.40 ± 0.01 g | 36.83 ± 0.88 de |

The values were the means of six replicates ± standard errors. Different letters within columns indicate that there were significant differences at a 0.05 level of probability.

### 3.2.3. Strawberry Yields

As shown in Table 5, fertilization treatments had a significant influence on all strawberry yield parameters in the 2019/2020 and 2020/2021 combined growing seasons. The T2 treatment (for early yield); T1, T2, and T3 (for total and marketable yield); T2, T3, T4, and T5 (for unmarketable yield); and T2 and T3 (for fruit firmness), were the most effective patterns in the studied seasons. Therefore, the 112.5 kg $K_2SO_4$ acre$^{-1}$ + 1000 mg L$^{-1}$ nano-chitosan-K (T2) and 112.5 kg $K_2SO_4$ acre$^{-1}$ + 750 mg L$^{-1}$ nano-chitosan-K (T3) treatments were the most established for enhancing strawberry yield characteristics across the two seasons studied.

**Table 5.** Effect of potassium treatments on yield traits of strawberry plants in the combined seasons of 2019/2020 and 2020/2021.

| Treatment | Early Yield (Ton Acre$^{-1}$) | Total Yield (Ton Acre$^{-1}$) | Marketable Yield (Ton Acre$^{-1}$) | Unmarketable Yield (Ton Acre$^{-1}$) | Fruit Firmness (g cm$^{-2}$) |
|---|---|---|---|---|---|
| T1 | 3.73 ± 0.01 b | 23.84 ± 0.61 a | 20.96 ± 0.45 a | 1.98 ± 0.06 b | 11.21 ± 0.37 bcd |
| T2 | 4.03 ± 0.02 a | 23.70 ± 0.89 a | 21.61 ± 0.27 a | 2.10 ± 0.04 ab | 12.03 ± 0.46 ab |
| T3 | 3.82 ± 0.03 b | 24.20 ± 0.62 a | 21.56 ± 0.15 a | 2.15 ± 0.02 a | 12.61 ± 0.17 a |
| T4 | 3.54 ± 0.01 c | 20.82 ± 0.56 bcd | 18.66 ± 0.17 bc | 2.08 ± 0.05 ab | 11.55 ± 0.68 abc |
| T5 | 3.45 ± 0.04 cd | 22.08 ± 0.74 b | 19.19 ± 0.38 b | 2.07 ± 0.04 ab | 11.21 ± 0.32 bcd |
| T6 | 2.74 ± 0.03 fg | 19.29 ± 0.70 e | 16.92 ± 0.14 d | 1.22 ± 0.08 d | 9.75 ± 0.21 e |
| T7 | 3.29 ± 0.16 d | 21.42 ± 0.11 bc | 19.47 ± 0.41 b | 1.39 ± 0.06 c | 10.58 ± 0.39 cde |
| T8 | 3.08 ± 0.16 e | 20.22 ± 0.57 cde | 18.66 ± 0.37 bc | 0.95 ± 0.01 e | 11.17 ± 0.39 bcd |
| T9 | 3.35 ± 0.01 cd | 20.00 ± 0.48 cde | 18.49 ± 0.19 bc | 1.02 ± 0.01 e | 10.82 ± 0.38 bcde |
| T10 | 2.91 ± 0.02 ef | 19.91 ± 0.63 de | 17.85 ± 0.53 cd | 0.99 ± 0.06 e | 10.19 ± 0.18 de |
| T11 | 2.62 ± 0.01 g | 16.44 ± 0.32 f | 14.33 ± 0.39 e | 0.95 ± 0.02 e | 9.99 ± 0.25 de |
| T12 | 3.03 ± 0.04 e | 19.85 ± 0.53 de | 17.50 ± 0.82 cd | 1.07 ± 0.02 e | 10.92 ± 0.38 bcde |

The values were the means of six replicates ± standard errors. Different letters within columns indicate that there were significant differences at a 0.05 level of probability.

### 3.2.4. Strawberry Fruit Quality

The beneficial effects of 112.5 kg $K_2SO_4$ acre$^{-1}$ + 1000 mg L$^{-1}$ nano-chitosan-K (T2) and 112.5 kg $K_2SO_4$ acre$^{-1}$ + 750 mg L$^{-1}$ nano-chitosan-K (T3) on strawberry fruit quality were also observed (Table 6). In this context, T2 (for total soluble solids); T1, T2, and T3 (for vitamin C); T2, T3, and T7 (for acidity); T1 and T2 (for total sugar); and T2, with a non-significant difference of T3 (for anthocyanin content), recorded significant increases and were the most effective practices with the highest yield increases.

**Table 6.** Effect of potassium treatments on fruit quality traits of strawberry plants in the combined seasons of 2019/2020 and 2020/2021.

| Treatment | Total Soluble Solids (%) | Vitamin C (mg 100 g$^{-1}$ FW) | Acidity (%) | Total Sugar (mg g$^{-1}$ FW) | Anthocyanin (mg 100 g$^{-1}$ FW) |
|---|---|---|---|---|---|
| T1 | 10.01 ± 0.02 bc | 53.22 ± 0.23 a | 1.71 ± 0.03 b | 6.98 ± 0.13 a | 67.85 ± 5.23 cd |
| T2 | 10.60 ± 0.02 a | 54.16 ± 0.61 a | 1.81 ± 0.01 a | 6.94 ± 0.03 ab | 77.68 ± 0.38 a |
| T3 | 10.26 ± 0.09 b | 54.08 ± 0.34 a | 1.76 ± 0.01 ab | 6.88 ± 0.03 abc | 75.34 ± 0.55 ab |
| T4 | 9.78 ± 0.05 c | 51.24 ± 0.45 b | 1.58 ± 0.03 c | 6.88 ± 0.02 abc | 71.15 ± 0.47 abc |
| T5 | 9.41 ± 0.19 d | 51.21 ± 0.12 b | 1.44 ± 0.01 d | 6.83 ± 0.01 bcd | 63.49 ± 5.40 d |
| T6 | 7.94 ± 0.09 h | 42.33 ± 0.33 f | 1.23 ± 0.05 f | 6.74 ± 0.02 d | 66.24 ± 0.89 cd |
| T7 | 9.86 ± 0.11 c | 52.00 ± 0.21 b | 1.76 ± 0.06 ab | 6.87 ± 0.01 abc | 70.30 ± 0.65 bcd |
| T8 | 9.36 ± 0.11 d | 47.88 ± 0.43 c | 1.37 ± 0.01 de | 6.77 ± 0.01 cd | 69.11 ± 0.80 bcd |
| T9 | 8.92 ± 0.04 e | 47.68 ± 0.21 c | 1.39 ± 0.01 de | 6.83 ± 0.02 bcd | 67.54 ± 0.17 cd |
| T10 | 8.55 ± 0.03 f | 46.19 ± 0.29 d | 1.33 ± 0.01 e | 6.84 ± 0.04 bcd | 66.93 ± 0.22 cd |
| T11 | 8.20 ± 0.02 g | 43.78 ± 0.38 e | 1.33 ± 0.03 e | 6.77 ± 0.02 cd | 67.61 ± 0.77 cd |
| T12 | 8.60 ± 0.10 f | 47.66 ± 0.33 c | 1.32 ± 0.02 e | 6.80 ± 0.01 cd | 68.67 ± 1.09 bcd |

The values were the means of six replicates ± standard errors. Different letters within columns indicate that there were significant differences at a 0.05 level of probability.

### 3.3. Principal Component Analysis

A biplot analysis was used to investigate the relationship between the treatments applied and the evaluated fruit yield and quality traits as an average of the 2019/2020 and 2020/2021 seasons (Figure 3).

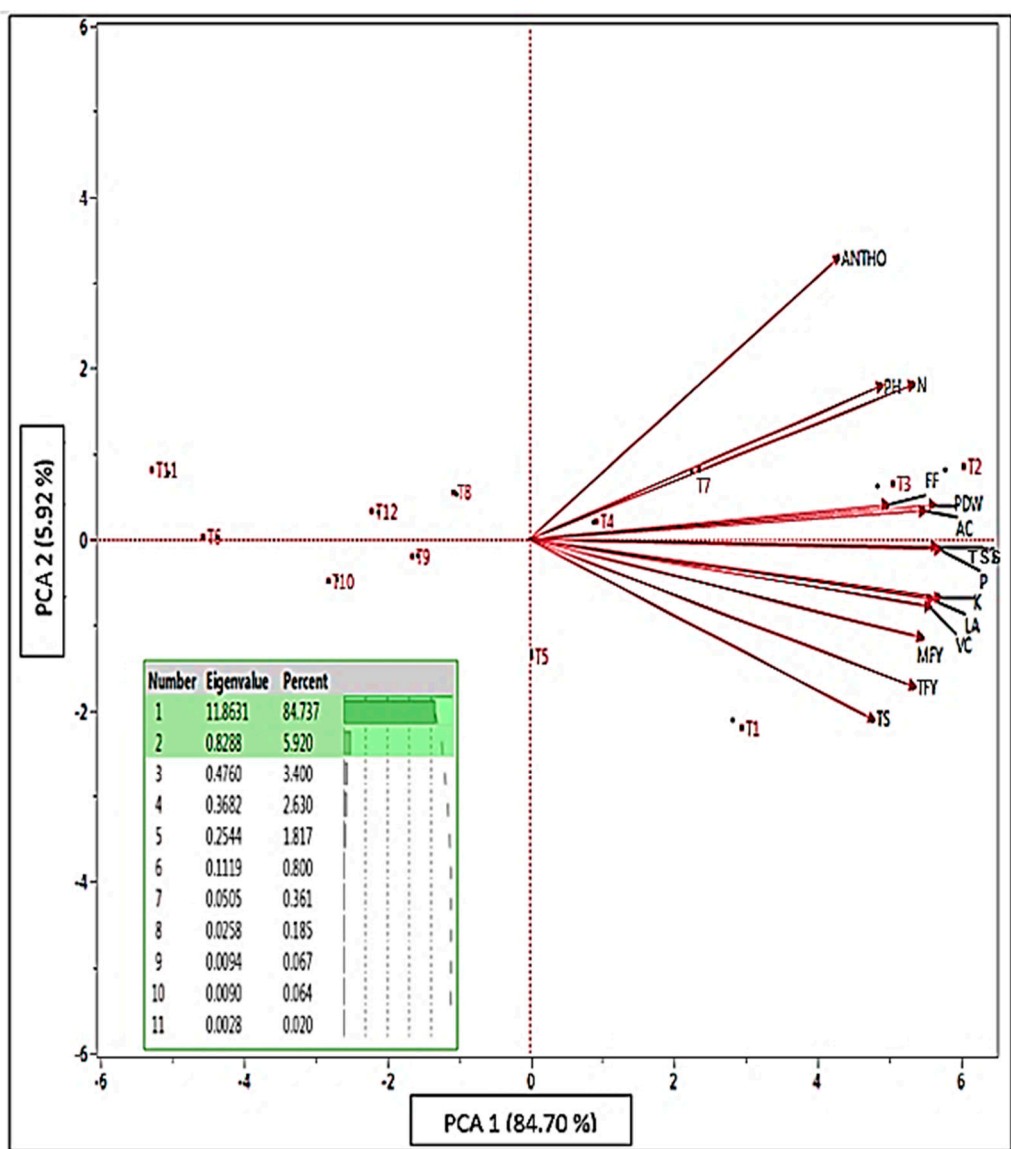

**Figure 3.** Vector view of the biplot depicting the interrelationship among studied treatments and traits of strawberry plants. ANTHO, anthocyanin; PH, plant height; N, nitrogen; FF, fruit firmness; PDW, plant dry weight; AC, acidity; TSS, total soluble solids; P, phosphorus; K, potassium; LA, leaf area; VC, vitamin C; MFY, marketable fruit yield; TFY, total fruit yield; TS, total sugar.

The lines emanating from the central point of each biplot indicate positive or negative correlations of different variables, whereas their proximity indicates the strength of the correlation with specific treatments. Strawberry marketable fruit yield (MFY) was strongly positively associated with all studied traits because the vector trait of MFY formed an acute angle of less than 90 degrees (<90°) with the vectors of these traits. The small acute angles between the vectors of PH, N, FF, PDW, AC, TSS, P, K, LA, VC, MFY, TFY, and TS were highly positively correlated. There was a near-zero correlation between ANTHO and TS due to the nearly perpendicular vectors (r = cos 90 = 0). The biplot model also explained 90.62% of the total variation in the standardized data. The first two principal components (PCA 1 and PCA 2) explained 84.70% and 5.92% of the variance, respectively. The polygon

view of the biplot distinguishes treatments with the highest values for one or more traits. In general, T2 and T3 were the most effective treatments for almost all studied strawberry traits, with high responses in N, P, and K (longer vectors). T6, T8, T9, T10, T11, and T12 were the least effective treatments for all estimated traits.

## 4. Discussion

Regarding the concentration of available K in the studied soil before cultivation, it reached 168 μg g$^{-1}$ (Table 1), which is considered to be medium, as reported by [33]. Many factors affect K dynamics in the soil such as the soil physio-chemical properties, soil microbial activities, and soil–plant interactions [13]. Therefore, it is necessary to add K fertilizers to replenish the depletion of K in soil fertility to meet a plant's requirements during its different growth physiological stages. In the combined two studied seasons of 2019/2020 and 2020/2021, it was found that increasing the soluble K, exchangeable K, and total K by applying 150.0 $K_2SO_4$ (100% of the recommended $K_2SO_4$) followed by 112.5 $K_2SO_4$ + 1000 mg L$^{-1}$ nano-chitosan-K (75% of the recommended $K_2SO_4$ combined with 1000 mg L$^{-1}$ nano-chitosan-K) increased soil K availability (Figure 2). $K_2SO_4$, a good solubility, chlorine-free K fertilizer, is the most effective way to increase soil K content [34]. However, applying 1000 mg L$^{-1}$ nano-chitosan-K without $K_2SO_4$ or adding 50% of the recommended dosage yielded the highest percentage of fixed K in sandy soil. Due to the decreased K fertilizer application in the soil, the concentration in this fraction was incorporated into the crystal lattice structure of clay minerals [13]. In this regard, [9] found that nano-K-sprayed wheat plants utilized the maximum amount of exchangeable K from sandy loam soil leading to the least amount of K loss by leaching. Nano-chitosan is a cationic biopolymer considered to be one of the most attractive highly reactive biopolymers mainly due to the presence of amino and hydroxyl functional groups on its backbone structure that embrace chemical linking with nutrients such as K [20,35]. The new nanocomposite works in the soil as a slow-release fertilizer, conserving K from leaching and increasing its bioavailability. Therefore, combining the two components of nano-chitosan and nano-K had a better effect on soil, and consequently on the cultivated plants [20].

The improvement of vegetative growth characteristics such as plant height, fresh and dry weights, numbers of leaves per plant, and leaf areas of strawberry plants with an increasing K fertilization rate either in soil or by spraying nano-chitosan-K on foliage was of interest (Table 3). The best results were with the combination of the two applications. This could be attributed to the increased uptake of K and its associated role in enzyme activation, protein synthesis, photosynthesis, osmoregulation, stomatal movement, energy transfer, phloem transport, cation–anion balance, and stress resistance [36–38]. Similarly, [9] proved that K had a stimulatory effect on the weight of wheat plants and better results were obtained when nanoformulation was applied as compared with conventional fertilizers.

Because the application of the T2 and T3 treatments caused an increase in the availability of nutrients, which, in turn, caused an improvement in the ability of plant roots and leaves to absorb nutrients, a high level of NPK and chlorophyll was found in the strawberry plant foliage (Table 4). Due to the fact that the nanoparticles had a high specific surface area and, consequently, a high reactive potential, higher absorption of nano-chitosan-K was achieved, and this resulted in a beneficial influence on plant growth [18]. Increases in both the NPK and chlorophyll content in plant foliage have been linked to the use of chitosan-K, which aided plant absorption of soil-water and nutrients, increased chlorophyll synthesis in plant leaves, and ultimately improved photosynthesis [39], which was reflected in the robust growth of plants.

Crop yields increased with increased plant growth and nutrient uptake. T2 and T3 improved the early fruit yield, marketable fruit yield, total fruit yield, and fruit firmness (Table 5). Crop productivity increased within the range from 6% to 17% with nanofertilizers. The ease with which nanofertilizers penetrated the stomata of leaves caused the increase (via a topical application). Since nutrient use efficacy is increased by at least 20% for most applied nutrients, nano-based fertilizers are much more effective than conventional

fertilizers [17]. Chitosan nanoparticles are easily absorbed by leaves and translocated to stems to boost plant growth and the yields of different crops [40].

Since both chitosan and K in the form of nanoparticles (Figure 1) are tiny in size, have high absorption capacity, and are dispersed and rapidly and optimally absorbed and taken up by plants [41], treating plants with nano-chitosan-K could effectively meet their nutrient needs [42]. The physical and chemical properties of nanoparticles that are small (1 to 200 nm) seem to be more effective [43,44].

Chitosan-K inhibits cell wall disintegration, slows the aging process, and decreases the formation of $H_2C=CH_2$ [36,45]. Thus, it is likely that the effectiveness of the T2 treatment was demonstrated by maintaining fruit firmness due to a reduction in ethylene formation. In addition, nano-chitosan has been shown by [46] to improve plant metabolic activity and to facilitate the transport of active chemicals across cell membranes. It also had positive effects on the productivity and quality of the plants. Because of its natural origin, non-toxicity, safety, and biodegradability, nano-chitosan has the potential to replace agrochemicals in the reduction of abiotic stresses [45]. Furthermore, K is a macronutrient that serves the same purpose in protecting plants from abiotic stresses [11]. Foliar application of nano-chitosan-K resulted in the increased accumulation of TSS, total sugar, and vitamin C in plant cells, which were osmotic adjustments and important indicators for fruit quality, in addition to fruit acidity and anthocyanin content which were responsible for fruit color [36,46] (Table 6). The potential role of chitosan in enhancing the availability and uptake of water and nutrients by regulating cell osmotic pressure and enzyme activities could explain its augmentative effect on strawberry growth and yield while improving fruit quality [47,48]. Additionally, the foliar application of chitosan has been shown to maintain the membrane stability of the leaf and to increase the levels of antioxidant enzymes in apple [49]. It has been demonstrated that chitosan improved sugar metabolism [50], thereby improving the quality of strawberry fruit juice. Similarly, a K supply induced stress tolerance with improvements in growth, chlorophyll synthesis, antioxidant enzyme activation, gas exchange traits, and sugar content [51]. In addition, K can play a crucial role in plant–water relations by encouraging water uptake by plants. This helps plants to achieve optimal turgor and membrane stability [52]. K is essential for photosynthesis because it facilitates translocation [53], modifies the osmotic charge [54,55], and improves crop growth, productivity, and quality [56].

The principal component analysis (PCA) biplots were useful for visualizing the relationship between treatments and the fruit production and quality characteristics of the strawberry plants that were tested (Figure 3). The first two principal components, as shown in the biplots, reflected more than 60% (90.62%) of the total variance. The utilized biplot model, therefore, provided a good fit. According to [57], the first two principal components should reflect more than 60% of the total variation in order to obtain a good fit for the biplot model. In addition, the ideal test trait should have the largest vector of all traits because it should be able to effectively differentiate between treatments [58]. The results that were obtained were consistent with those that were reported by [59,60] concerning the effect of nanofertilizers, including chitosan nanoparticles, on the quality and productivity of strawberries. As a result, the use of exogenously applied beneficial compounds in a nano form, such as chitosan and K, achieved favorable progress in plant nutrition, resulting in high quantity and a high quality of crop products.

## 5. Conclusions

By utilizing the advantages of nanotechnology, it is possible to reduce the amount of K used in strawberry fertilization. Yields of high-quality strawberries were achieved by spraying plants with a solution containing nano-sized chitosan loaded with nano-K and by adding K fertilizer to the soil at a rate of 75% of the recommended dosage (the recommended dosage of applying K fertilizers to the sandy soil for strawberry cultivation is 150.0 kg $K_2SO_4$ acre$^{-1}$). Furthermore, applying 75.0 kg $K_2SO_4$ acre$^{-1}$ (50% of the recommended dosage) in the soil plus spraying the plants with 1000 mg L$^{-1}$ nano-chitosan-K (T7)

gave high and promising results. Thus, accelerating plant growth and productivity through the application of nano-chitosan-K can open up new perspectives in agricultural practices, as it is of natural origin, non-toxic, safe, biodegradable, and an excellent alternative to agrochemicals for achieving sustainability in the agricultural sector. However, additional field research is required to determine the safety of nano-treated plants for human consumption and to assess the effects of nano-chitosan-K on various pathways of metabolism in strawberries. In addition, technical and economic studies related to different applications should be conducted for comparisons.

**Author Contributions:** Conceptualization, S.H.A.-E., Y.A.E.-G.E.-G., F.A.H. and N.M.T.; Data curation, S.H.A.-E., F.A.H., M.F.M.I., E.I.E.-H., Z.K.A., S.M.A.-Q. and N.M.T.; Formal analysis, M.F.M.I., Z.K.A., D.B.E.D. and N.A.A.-H.; Funding acquisition, E.I.E.-H.; Investigation, S.H.A.-E., Y.A.E.-G.E.-G., F.A.H. and N.M.T.; Methodology, S.H.A.-E., Y.A.E.-G.E.-G., F.A.H. and N.M.T.; Project administration, E.I.E.-H.; Resources, Y.A.E.-G.E.-G., E.I.E.-H., Z.K.A., N.A.A.-H., S.M.A.-Q. and N.M.T.; Software, Z.K.A., D.B.E.D., N.A.A.-H. and S.M.A.-Q.; Supervision, S.H.A.-E. and M.F.M.I.; Validation, Y.A.E.-G.E.-G., M.F.M.I., Z.K.A., D.B.E.D. and N.M.T.; Visualization, S.H.A.-E., F.A.H., D.B.E.D., N.A.A.-H. and S.M.A.-Q.; Writing—original draft, S.H.A.-E., Y.A.E.-G.E.-G., F.A.H. and N.M.T.; Writing—review and editing, S.H.A.-E., M.F.M.I., E.I.E.-H., Z.K.A., D.B.E.D. and S.M.A.-Q. All authors have read and agreed to the published version of the manuscript.

**Funding:** This work was funded by the Deanship of Scientific Research, Taif University.

**Institutional Review Board Statement:** Not applicable.

**Informed Consent Statement:** Not applicable.

**Data Availability Statement:** Not applicable.

**Acknowledgments:** The authors would like to acknowledge the Deanship of Scientific Research, Taif University for funding this work.

**Conflicts of Interest:** The authors declare no conflict of interest.

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
