# Peer review of "Influence of Nano-Chitosan Loaded with Potassium on Potassium Fractionation in Sandy Soil and Strawberry Productivity and Quality"

_agronomy, doi:10.3390/agronomy13041126_

Round 1

Reviewer 1 Report

This article presented Influence of Nano Chitosan–Loaded K on Potassium Fractionation in Sandy Soil and Strawberry Productivity and Quality. The study is well organized and data is well arranged. The findings would be helpful for future studies. Before recommending this article for publication, there are some shortcomings for that should be resolve.

In abstract what does mean by alone or combination. The sentence is confusing should be revise.

Also briefly describe other methods in abstract

Results of chitosan NPs are not presented in the abstract.

Line 60 should be cited with recent study https://doi.org/10.1002/aoc.5190

Line 70, also add fruit protection, food packaging etc. and should be cited with recent study https://doi.org/10.1016/j.bcab.2020.101729

Line 72 lack reference.

Literature review of the effect of Chitosan based NPs must be updated and cover other aspects as well specifically in agriculture sector. How combination of chitosan with combination of other nutrients or metals work efficiently.

Why chitosan alone is not sufficient.

Why combination K was required.

Chitosan-K NPs are not characterized in this study. Provide the reason

Conclusion is well presented. However, future recommendations based on the obtained results must be added in the conclusion section.

Author Response

Dear reviewer I

Thank you so much for your precious comments and your constructive opinions; we acknowledge your time and effort in revising our manuscript. We carefully revised the manuscript and made the point-to-point response to each comment or suggestion. Hopefully, the current revised version of our MS is satisfactory now to meet your high expectations.

Reviewer 2 Report

Unfortunately the characterization of Nano Chitosan–Loaded K formation is not enough to confirm the formation of nanocomposite 

Author Response

Dear reviewer II

Thank you so much for your precious comments. We acknowledge your time and effort in revising our manuscript. We carefully revised the manuscript and made the point-to-point response to each comment or suggestion. Hopefully, the current revised version of our MS is satisfactory now to meet your high expectations.

Reviewer 3 Report

Abstract : Please address K-related issues with respect to strawberry grown on  sandy soils to straightway hit the issue. How did you establish your experimental soil was low in available supply of K. to obtain such crop response. How do you justify such combination of K carrying 112.5 kg K2SO4 acre−1+1000 mg L−1 nano chitosan–K. Conclusive statement needs to be modified , since you need other nutrients as well to be applied with other nutrients as well to optimise the crop response of applied K. 

Introduction :Authors must highlight the role of nano-K versus conventional K-fertilizers. And the hypothesis of combination of nano-K plus conventional fertilizers seems quite verbose , needs to be relooked  in the light of saving on use of K-fertilizers through use of nano-K.

Materials and methods: How do you justify available -K of 168 ug/g soil is likely to be studied for K-response , we need a strong justification ( Table 1). We need to depict the complete irrigation schedule ( Line 122-123). Such soils are likely to be deficient of Fe and Zn , why authors have ignored such important nutrients while studying such crop responses.  Difficult to understand  the relevance of these lines ( Line 138-142). Not so clear , how did you identify nano-K particles ( Fig. 1). 

Results: Authors need to highlight different K-fractions for conventional K-fertilizers versus chitosan-K , otherwise it looks quite ordinary presentation ( Line 200-206 ) . Authors , please try to portray your results using pooled data analysis of two seasons , instead of talking year-wise , unless results are varying over years ( Line 209-220 ). A unified Fig2 should be provided , not two different figures, unnecessary it lengthens the manuscript without any additional quality and same apply for Table 2 , Table 3, Table 4 , and Table 5  ( nothing can be interpreted with this kind of  presentation ) , please . This is very poor style of presentation , since results are not varying over years. Please coin the treatments and use those coined treatment symbols , otherwise every time explaining those treatments consumes lot of space in the manuscript ( Fig. 3). Discussion is really weak.  

Conclusion : I do not see any advantage of combining conventional K-fertilizer with chitosan-K, unless authors undertake massive revision to re-write the whole manuscript on suggested lines. Manuscript needs complete over-hauling and then further review  to arrive at any decision . 

Author Response

Dear reviewer III

Thank you so much for your precious comments and suggestions. We acknowledge your time and effort in revising our manuscript. We carefully revised the manuscript and made the point-to-point response to each comment or suggestion. Hopefully, the current revised version of our MS is satisfactory now to meet your high expectations.

Reviewer 4 Report

The work concerns a very current topic, which is the optimization of strawberry fertilization. It proves the effectiveness of using a reduced dose of mineral fertilizer in combination with foliar application of nano chitosan loaded K. This makes it the most suitable for publication in Agronomy, in Special Issue: "Effects of Soil Fertility and Plant Growth Promoters on Growth, Yield, and Quality of Crops". Each chapter has been prepared with great care. The proposed literature is current and very relevant.

Author Response

Dear reviewer IV

Thank you so much for your precious comments and your constructive opinions; we acknowledge your time and effort in revising our manuscript. We carefully revised the manuscript and made the point-to-point response to each comment or suggestion. Hopefully, the current revised version of our MS is satisfactory now to meet your high expectations.

Reviewer 5 Report

The paper is interesting. The authors tested three rates of potassium sulfate fertilization and foliar application of four doses of nano chitosan-K in strawberries. However, I have some comments on the manuscript. In the introduction, reference 2 refers to the impact of drought on photosynthesis. The authors should consider changing this reference to one that refers to the use of chemical fertilizers and pesticides in strawberries. In the methodology section, the authors do not provide sufficient details on the synthesis of nano chitosan-K. Additionally, the treatment used as a control in the experiment is not clear. In the results, the authors mention that treatments T1, T5, and T6 were the most effective for nitrogen content in the first season. However, Table 3 shows that treatments T3 and T7 were the most effective. The results presented for fruit quality are not entirely clear, specifically for anthocyanin in the second season. The authors report that T8 stands out, but it shares similar results with treatments from T6 to T12

Author Response

Dear reviewer V

Thank you so much for your precious comments and suggestions; we acknowledge your time and effort in revising our manuscript. We carefully revised the manuscript and made the point-to-point response to each comment or suggestion. Hopefully, the current revised version of our MS is satisfactory now to meet your high expectations.

Round 2

Reviewer 2 Report

Figure 1. not clear 

where other characterization 

Author Response

Dear reviewer II

Thank you so much for your valuable comments. We have made the corrections and follow your instructions as closely as possible. Hopefully, these corrections can meet now your high expectations.

please see the attached file of our responses 

Thank you so much

Reviewer 3 Report

Abstract  : Please revise the hometake message ( Line 51-53) , this is too big claim , ignoring the role of loading nano -K with chitosan . 

Introduction :  Please do not use repetitive statements ( Line 74-75) versus ( Line 86-87) .

Materials and methods: Please revisit the soil moisture content at field capacity with available water capacity of 15% , its too high. Where is the statement that you addressed the other nutrients while experimenting the response of nano-K.?. Please take care that whatever you write as your response against the reviewers comments  , those must figure in your revision, instead of only wordly statement .  Please add the quantity of water added to maintain the soil moisture content at field capacity ( Line 165-167).  

Results : Please provide the pooled data analysis of two seasons , no need to highlight seasonwise data for Table 2-Table5. 

Discussion : First thing we need to justify ( reasons for crop response)  the response of chitosan-K at 168 ug/kg  in sandy soil , again authors must bring discussion in text form , rather simply replying for reviewers comments  .Do you feel combination treatment brings two complimenting kinetics of nutrient release , thereby, ensuring the crop response. 

Conclusion : What do you mean by 75% of K of RDF through chitosan-K, is it not a high dose , think of from farm application point of view, unless cost -benefit is worked out.   

Author Response

Dear reviewer III
Thank you so much for your valuable comments. We have made the corrections and follow your instructions as closely as possible. Hopefully, these corrections can meet now your high expectations.
please see the attached file of our responses 
Thank you so much
